# Muscle Proteomic Profile before and after Enzyme Replacement Therapy in Late-Onset Pompe Disease

**DOI:** 10.3390/ijms22062850

**Published:** 2021-03-11

**Authors:** Manuela Moriggi, Daniele Capitanio, Enrica Torretta, Pietro Barbacini, Cinzia Bragato, Patrizia Sartori, Maurizio Moggio, Lorenzo Maggi, Marina Mora, Cecilia Gelfi

**Affiliations:** 1Gastroenterology and Digestive Endoscopy Unit, IRCCS Policlinico San Donato, 20097 Milano, Italy; Manuela.Moriggi@grupposandonato.it; 2Neuromuscular Diseases and Neuroimmunology Unit, Fondazione IRCCS Istituto Neurologico Carlo Besta, 20133 Milano, Italy; cinzia.bragato@istituto-besta.it (C.B.); Lorenzo.Maggi@istituto-besta.it (L.M.); moramarinad@gmail.com (M.M.); 3Department of Biomedical Sciences for Health, University of Milan, 20090 Milano, Italy; daniele.capitanio@unimi.it (D.C.); pietro.barbacini@unimi.it (P.B.); patrizia.sartori@unimi.it (P.S.); 4IRCCS Istituto Ortopedico Galeazzi, 20161 Milano, Italy; enrica.torretta@grupposandonato.it; 5Department of Medicine and Surgery, PhD Program in Neuroscience, University of Milano-Bicocca, 20126 Milano, Italy; 6Neuromuscular and Rare Diseases Unit, Department of Neuroscience, Fondazione IRCCS Ca’ Granda, Ospedale Maggiore Policlinico, 20122 Milano, Italy; maurizio.moggio@unimi.it

**Keywords:** pompe disease, sarcopenia, autophagy, rare disease, proteomics, mass spectrometry

## Abstract

Mutations in the acidic alpha-glucosidase (GAA) coding gene cause Pompe disease. Late-onset Pompe disease (LOPD) is characterized by progressive proximal and axial muscle weakness and atrophy, causing respiratory failure. Enzyme replacement therapy (ERT), based on recombinant human GAA infusions, is the only available treatment; however, the efficacy of ERT is variable. Here we address the question whether proteins at variance in LOPD muscle of patients before and after 1 year of ERT, compared withhealthy age-matched subjects (CTR), reveal a specific signature. Proteins extracted from skeletal muscle of LOPD patients and CTR were analyzed by combining gel based (two-dimensional difference gel electrophoresis) and label-free (liquid chromatography-mass spectrometry) proteomic approaches, and ingenuity pathway analysis. Upstream regulators targeting autophagy and lysosomal tethering were assessed by immunoblotting. 178 proteins were changed in abundance in LOPD patients, 47 of them recovered normal level after ERT. Defects in oxidative metabolism, muscle contractile protein regulation, cytoskeletal rearrangement, and membrane reorganization persisted. Metabolic changes, ER stress and UPR (unfolded protein response) contribute to muscle proteostasis dysregulation with active membrane remodeling (high levels of LC3BII/LC3BI) and accumulation of p62, suggesting imbalance in the autophagic process. Active lysosome biogenesis characterizes both LOPD PRE and POST, unparalleled by molecules involved in lysosome tethering (VAMP8, SNAP29, STX17, and GORASP2) and BNIP3. In conclusion this study reveals a specific signature that suggests ERT prolongation and molecular targets to ameliorate patient’s outcome.

## 1. Introduction

Pompe disease is a lysosomal storage disorder caused by mutations in the gene coding for acid alpha-glucosidase (GAA). This enzyme is responsible for glycogen breakdown in lysosomes. Striated muscle is the primary target and patients undergo cardiac and muscle myopathy due to glycogen accumulation. The disease clinical course depends on specific mutations, levels of mutated GAA, and residual activity of the enzyme [1]. Late-onset Pompe disease (LOPD) is characterized by up to ∼20% normal residual GAA activity, and by progressive general proximal and axial muscle weakness and atrophy, leading to respiratory failure, ambulation deficiency, and/or hyperCKemia without cardiac involvement in the majority of patients [2,3]. The degree of deficiencies correlates with disease duration, increasing with time the need for ventilation support and wheelchair dependency [4]. Enzyme replacement therapy (ERT), based on repeated intravenous infusions of recombinant human GAA (hGAA), represents, at present, the only available treatment for Pompe disease. Upon more than 15 years from its launch, although the response of LOPD patients to ERT shows a high degree of heterogeneity, results indicate that ERT can stabilize disease progression. After the initial improvement, most LOPD adult patients maintain a plateau, which may last for several years, while in a subset of patients, after an initial positive response to ERT, secondary deterioration may occur after 1–2 up to 7–8 years of treatment [5]. The treatment has a positive effect on muscle strength, motor function, respiratory measurements, survival, decreases the need of invasive and non-invasive ventilation and ambulatory support [6,7,8]. However, a recent study indicates that ERT was able to improve respiratory function within the first few months, followed by a gradual return to baseline [8]. In muscles, ERT efficacy in improving muscle symptoms is variable and independent of disease duration and extent of enzyme deficiency [8,9].

The severe decrement of GAA activity causes accumulation of undigested glycogen and rupture of lysosomes and has long been considered the mechanism through which progressive and persisting muscle damage takes place. However, recent studies have shown that the pathogenetic machinery is more complex and implies autophagy impairment, accumulation of potentially toxic undegraded constituents and oxidative stress [10]. These observations have stimulated a reassessment of the disease pathogenetic mechanisms and a search for new therapeutic approaches, including overexpression of transcription factor EB, adeno-associated virus (AAV)-mediated gene therapy and genetic manipulation of autophagy [11,12,13,14,15,16,17,18].

In spite of the progress reached thus far, several questions remain open, in particular the effect exerted by metabolic changes at muscle level and the signaling associated with lysosome and autophagosome tethering. Recent discoveries indicate lysosomes to be central to the mammalian target of rapamycin complex 1 (mTORC1) pathway, and relate these organelles to cell growth and metabolism through mechanisms based on signals from nutrients, energy and growth factors, which play a fundamental role in mTORC1 activity and contribute to blunt macroautophagy [19,20,21]. Concerning glycogen, it provides energy to the skeletal muscles during high-intensity exercise and is metabolized in the cytosol under the control of the phosphorylated form of glycogen phosphorylase, Pi and Ca^2+,^ and in autophagic vacuoles [22,23]. In the latter, glycogen is degraded into non-phosphorylated glucose by enzymes located in lysosomes, including the glycogen-hydrolyzing GAA, synthesized and mannosylated in the rough endoplasmic reticulum (ER). The glycosylation and proper folding of GAA in the ER are crucial for transport to the Golgi where the enzyme acquires the targeting signal necessary for binding to the mannose 6-phosphate receptor. The enzyme is then delivered to the lysosome where, before entering, it is proteolytically cleaved at both the amino- and carboxyl-terminal ends [24]. Defects in GAA cause lysosomal dysfunction due to glycogen overload, thus compromising cell homeostasis [25].

The overall aim of the present study is to precisely compare characteristics of the muscle proteome of pre-ERT and post-ERT patient muscle, versus healthy age-matched control muscle, so as to understand why clinical efficacy of ERT decreases over time. In addition, this study aims at identifying molecules involved in signaling events that generate muscle weakness and lysosome dysfunction, thus providing a picture of molecular players involved in the derangement of muscle metabolism and structure in LOPD.

To precisely decipher the effects of ERT on muscle tissue, proteomic analyses have been adopted based both on our experience in proteomics of skeletal muscle from patients and healthy subjects under various conditions [26,27,28,29,30,31,32], and on literature reports [33,34]. More specifically, a double proteomic approach has been applied based on two-dimensional difference gel electrophoresis (2D-DIGE) and label-free liquid chromatography-mass spectrometry (LC-MS/MS) analyses to determine both differential abundance of intact proteoforms and of trypsin digested proteins in muscle extracts of LOPD patients before and after 12 months of ERT. To corroborate proteomic results, ingenuity pathway analysis (IPA) has been carried out on the proteomic datasets. To provide a mechanistic understanding of events associated with hGAA response results have been validated on specific targets. 

## 2. Results

### 2.1. Electron Microscopy

Electron microscopy images of three samples, analyzed in the present study from which the material was still available, are shown in Figure 1 and Figure 2. Figures show muscle biopsies of the same LOPD patient before (LOPD PRE) and after ERT (LOPD POST). LOPD PRE muscle is characterized by glycogen overload in lysosomes (see close-up of Figure 1A), disorganization of the mitochondrial structure, derangement of myofibrils with disrupted sarcomeres, and disorganization of triad junction. LOPD POST muscle is characterized by recovery of lysosomal glycogen accumulation, indicating the positive effect of ERT on lysosome dysfunction. A significant number of glycogen granules dispersed between fibers (indicated by arrows in Figure 1B) and few mitochondria with visible cristae are observed. Notably, M and Z-line disruption and triad junction alteration persist in the post ERT muscle.

### 2.2. Comparative Proteomic Analysis 

Gel based (2D-DIGE) and label free (LC-MS/MS) proteomic approaches, were utilized to identify differentially expressed proteins in muscle tissues from LOPD PRE and LOPD POST patients compared to controls (CTR) (Appendix A, 2D-DIGE and label-free LC–MS/MS experimental design; Appendix A, Changed proteins in 2D-DIGE analysis; Appendix A, Changed proteins in label-free LC-MS/MS analysis). Overall, the abundance of 178 out of 900 identified proteins/proteoforms was altered in the muscle tissue of LOPD patients and 47 recovered to normal level after 1 year of ERT therapy. Proteins differentially abundant were grouped according to their function and listed in Appendix A.

### 2.3. Pathway Analysis

To predict biochemical pathways and biological functional processes associated with differentially expressed proteins in muscle extracts of LOPD PRE vs. CTR, LOPD POST vs. CTR, and LOPD PRE vs. LOPD POST, datasets were processed utilizing the ingenuity pathway analysis (IPA) software. This analysis allows identifying canonical pathways, down-stream effectors, and up-stream regulators predicted to be involved in LOPD and in the response to 1 year of ERT therapy in muscle tissue. 

#### 2.3.1. Canonical Pathways

The canonical pathway analysis enables recognition of key signaling pathways associated with differentially expressed proteins obtained by proteomic analysis. A total of 111 pathways were significantly associated with our data sets. Among those, 101 were significantly associated with LOPD PRE vs. CTR, 69 were significantly associated with LOPD POST vs. CTR, and 39 significantly associated with LOPD POST vs. LOPD PRE (Fisher’s right tailed exact test *p*-value < 0.05; Appendix A). 

The canonical pathway heatmap shown in Figure 3 displays the most significant results across datasets in LOPD PRE vs. CTR, in LOPD POST vs. CTR, and in LOPD POST vs. LOPD PRE. Results according to *p*-value are shown in panel A, and results according to z-scores are shown in panel B and in Appendix A. The orange and blue colors indicate, respectively, the predicted pathway activation or inhibition via the z-score statistic (z-score ≥ 2, ≤ −2). As shown in Figure 3B, the main pathways inhibited in LOPD PRE compared to CTR were oxidative phosphorylation and BAG2 signaling pathway, whereas sirtuin signaling, glycolysis, and PKR role in interferon induction and antiviral response were activated. In LOPD POST vs. CTR, oxidative phosphorylation and, to a lesser extent, necroptosis signaling pathway were inhibited, whereas sirtuin signaling, RhoA signaling, and role of PKR in interferon induction and antiviral response were activated. Moreover, in LOPD POST vs. LOPD PRE glycolysis and gluconeogenesis were inhibited.

The gene heatmap shown in Figure 4 specifically indicates levels of proteins associated with canonical pathway dysregulation described in Figure 3B by z-score statistics. Green and red colors refer, respectively, to decrease or increase for each individual protein statistically changed in our proteomics dataset. Grey color indicates statistically non-significant changes.

The following proteins of the oxidative phosphorylation pathway (4A) were increased in LOPD PRE compared to controls: NADH ubiquinone oxidoreductase subunit A9 (NDUFA9), ATP synthase membrane subunit f (ATP5MF), and cytochrome c oxidase subunit 6B1 (COX6B1). After ERT, ATP5MF, and COX6B1 were fully renormalized, whereas NDUFA9 was further increased compared to controls. When comparing LOPD PRE vs. controls, the following proteins were decreased: NADH ubiquinone oxidoreductase subunit A13 (NDUFA13), ubiquinol-cytochrome c reductase, Rieske iron-sulfur polypeptide 1 (UQCRFS1), NDUFA4 mitochondrial complex associated (NDUFA4), ATP synthase F1 subunit delta (ATP5F1D), NADH ubiquinone oxidoreductase core subunit S1 (NDUFS1), ubiquinol-cytochrome c reductase core protein 1 (UQCRC1), ATP synthase F1 subunit beta (ATP5F1B), ATP synthase F1 subunit alpha (ATP5F1A), ATP synthase F1 subunit gamma (ATP5F1C), cytochrome c oxidase subunit 4I1 (COX4I1), cytochrome c oxidase subunit 5A (COX5A), ATP synthase peripheral stalk subunit OSCP (ATP5PO), ubiquinol-cytochrome c reductase core protein 2 (UQCRC2), NADH ubiquinone oxidoreductase core subunit V1 (NDUFV1), and ATP synthase peripheral stalk subunit d (ATP5PD). After ERT, NDUFA13, NDUFS1, UQCRC2, and ATP5F1B were partially renormalized (i.e., their levels moved closer to, but remained significantly different from controls); the remaining proteins were further decreased, compared to the controls. Moreover, ATP synthase membrane subunit g (ATP5MG) was decreased in LOPD POST compared to controls, and ATP5PD was decreased in LOPD POST vs. LOPD PRE.

The following proteins of the sirtuin signaling pathway (4B) were up-regulated in LOPD PRE compared to controls: forkhead box O3 (FOXO3), protein kinase AMP-activated catalytic subunit alpha 1 (PRKAA1), NDUFA9, tubulin alpha 4a (TUBA4A), microtubule associated protein 1 light chain 3 beta(MAP1LC3B), phosphoglycerate kinase 1 (PGK1), phosphoglycerate mutase 2 (PGAM2), and lactate dehydrogenase A (LDHA); whereas, the following proteins were decreased: phosphofructokinase, muscle (PFKM), NDUFA13, UQCRFS1, voltage dependent anion channel 1 (VDAC1), lactate dehydrogenase B (LDHB), NDUFA4, NDUFS1, solute carrier family 25 member 4 (SLC25A4), ATP5F1B, ATP5F1A, ATP5F1C, pyruvate dehydrogenase E1 subunit alpha 1 (PDHA1), (VDAC2) voltage dependent anion channel 2, UQCRC2, (VDAC3) voltage dependent anion channel 3, glutamic-oxaloacetic transaminase 2 (GOT2), NDUFV1 and ATP5F1D. After ERT, levels of the proteins ATP5F1B, MAP1LC3B, NDUFA13, NDUFS1, PFKM, PGK1, UQCRC2, and VDACs showed a tendency toward control levels (i.e., values moved closer to, but remained significantly different from controls); whereas for all other proteins the discrepancy in respect to controls was increased.

The following proteins of the RhoA signaling pathway (4C) were increased in LOPD PRE compared to controls: skeletal muscle actin alpha 1 (ACTA1), myosin light chain phosphorylatable fast skeletal muscle (MYLPF), actin gamma 2 smooth muscle (ACTG2), and myosin light chain 1 (MYL1); whereas, titin (TTN) and myosin light chain 2 (MYL2) were decreased. After ERT, MYL1, and TTN remained dysregulated in LOPD POST compared to controls, although with a tendency to recover; whereas, ACTA1 level was increased compared to controls. MYLPF, ACTG2, and MYL2 were renormalized by ERT. Myosin light chain 6B (MYL6B) increased in LOPD POST compared to both control and LOPD PRE. ACTG2, MYL1, and cofilin 2 (CFL2) decreased in LOPD POST vs. LOPD PRE.

The following proteins of glycolysis (4D) and gluconeogenesis (4F) metabolism were increased in LOPD PRE compared to controls: fructose-bisphosphatase 2 (FBP2), enolase 3 (ENO3), PGK1, glyceraldehyde-3-phosphate dehydrogenase (GAPDH), PGAM2, aldolase fructose-bisphosphate A (ALDOA), and triosephosphate isomerase 1 (TPI); whereas PFKM, malate dehydrogenase 2 (MDH2) and malate dehydrogenase 1 (MDH1) were decreased. After ERT, GAPDH, PGAM2, ALDOA, TPI1 were renormalized; while FBP2, ENO3, PGK1, and PFKM remained dysregulated, although showing trends toward normalization; MDH1 and MDH2 were further decreased compared to control levels. Pyruvate kinase M1/2 (PKM) was decreased in LOPD POST vs. controls; whereas PGK1, PGAM2, TPI1, PKM, MDH1, and MDH2 were decreased in LOPD POST vs. LOPD PRE.

Concerning proteins involved in the role of PKR in interferon induction and antiviral response (4E) and in BAG2 signaling pathway (4H), the following proteins were decreased in LOPD PRE compared to controls: heat shock protein family A (Hsp70) member 5 (HSPA5), annexin A2 (ANXA2), heat shock protein family A (Hsp70) member 1A (HSPA1A/HSPA1B), heat shock protein family A (Hsp70) member 8 (HSPA8), heat shock protein 90 alpha family class A member 1 (HSP90AA1), and heat shock protein 90 alpha family class B member 1 (HSP90AB1). After ERT, HSPA8 and HSP90B1 were partially renormalized; whereas, HSPA1A/HSPA1B and HSP90AA1 were further decreased compared to control levels.

The following proteins of the necroptosis signaling pathway (4G) were decreased in LOPD PRE compared to controls: VDAC1, SLC25A4, VDAC2, and VDAC3, whereas glutamate-ammonia ligase (GLUL) was increased. After ERT, VDACs were only partially renormalized; whereas SLC25A4 was further decreased compared to controls.

#### 2.3.2. Diseases and Bio Functions

IPA analysis enables also to predict biological functional processes and disorders associated with differentially expressed proteins. Among a long list of disease and biofunctions the most statistically relevant as decreased functions, based on *p*-values were: sliding of filaments, hereditary myopathy, and muscle contraction. The full list is provided as Appendix A. 

In Figure 5 the disease and bio function heat map displays the most significant results, based on *p*-value and fold change, across datasets related to sliding of filaments (5B), hereditary myopathy (5C), and muscle contraction (5D).

More specifically, in sliding of myofilaments (5B) and in muscle contraction (5D), the following proteins were increased in LOPD PRE vs. CTR: troponin I2, fast skeletal type (TNNI2), troponin C2 fast skeletal type (TNNC2), actinin alpha 3 (gene/pseudogene) (ACTN3), ACTA1, troponin T3 fast skeletal type (TNNT3), tropomyosin 1 (TPM1), tropomyosin 2 (TPM2), MYL1, dysferlin (DYSF), tripartite motif containing 72 (TRIM72), myosin light chain, MYLPF, ACTG2, PGAM2, and ALDOA. Whereas, the following proteins were decreased in LOPD PRE vs. CTR: nebulin (NEB), titin-cap (TCAP), TTN, myosin binding protein C1 (MYBPC1), tropomyosin 3 (TPM3), MYL2, myosin heavy chain 7 (MYH7), troponin I1 slow skeletal type (TNNI1), myosin heavy chain 2 (MYH2), myosin heavy chain 4 (MYH4), and creatine kinase mitochondrial 2 (CKMT2). After ERT, proteomic analyses indicated that TNNI2, TNNC2, TNNT3, TPM1, TPM2, MYL1, NEB, TTN, MYBPC1, MYH7, and TRIM72 remained dysregulated in LOPD POST compared to controls, despite a tendency to recover; whereas, ACTN3, ACTA1, TPM3, DYSF, MYH2, MYH4, and CKMT2 increased their dysregulation compared to controls; whereas TCAP, MYL2, TNNI1, MYLPF, ACTG2, PGAM2, and ALDOA were renormalized. MYL6B and myosin heavy chain 3 (MYH3) were significantly differentially expressed only comparing LOPD POST vs. controls. Dystrophin (DMD) was up-regulated, whereas cysteine and glycine rich protein 3 (CSRP3) was down-regulated in LOPD POST only compared to LOPD PRE.

### 2.4. ER Stress

Dysregulation of proteins involved in the unfolded protein response (UPR) and increased levels of transitional endoplasmic reticulum ATPase (VCP) necessary for export of misfolded proteins from ER to the cytoplasm, and of HSPA5, suggest activation of the ER stress response. To corroborate these findings, UPR response was confirmed by immunoblotting of C/EBP-homologous protein (CHOP). Results showed increased levels of CHOP, particularly in LOPD POST compared to controls, indicating that the transcription factor is active and promotes UPR (Figure 6). 

### 2.5. Lysosomal Autophagy and Apoptosis

To correlate proteomic and IPA results with molecules involved in macroautophagy and apoptosis, several proteins were quantified by immunoblotting on muscle extracts in LOPD PRE and LOPD POST vs. controls. Specifically, Figure 7 shows the semi-quantitative results of microtubule associated protein 1 light chain 3 beta (LC3B), ubiquitin-binding protein p62 (p62), sintaxin 17 (STX17), synaptosomal-associated protein 29 kDa (SNAP29), endobrevin (VAMP8), Golgi reassembly-stacking protein 2 (GORASP2), lysosome-associated membrane protein 2 (LAMP2), BCL2/adenovirus E1B 19 kDa protein-interacting protein 3 (BNIP3). Figure 7b shows results of protein kinase AMP-activated catalytic subunit alpha 1 (AMPK), p38 mitogen-activated protein kinases 11 (p38β), and fork head box O3 (FoxO3). Results indicate that LC3BII/LC3BI, STX17, and LAMP2 followed the same behavior, all being increased in LOPD PRE and such increase was retained in LOPD POST compared to controls; while p62 and SNAP29 were increased in LOPD PRE compared to controls. VAMP8 and GORASP2 were at variance: VAMP8 decreased in LOPD POST whereas GORASP2 increased in LOPD POST compared to controls. Notably, regarding BNIP3, both monomer and dimer were increased in LOPD PRE compared to the controls, confirming dysregulation of the sirtuin pathway and its possible association to apoptosis under p38β-FOXO-BNIP3 axis. Increase of phospho-AMPK and phospho-p38β was observed in both LOPD PRE and LOPD POST compared to controls. AMPK increased both in LOPD PRE and POST compared to controls, whereas p38β was unchanged. Phospho-FoxO3 increased in LOPD PRE compared to POST and also compared to the controls. Meanwhile, FoxO3 was more abundant both in LOPD PRE and LOPD POST, compared to controls.

## 3. Discussion

Our analysis of muscle biopsies before (LOPD PRE) and after one year of ERT (LOPD POST) compared to age-matched controls, has allowed us to compare and contrast dysregulated proteins in the same patient. The aim was to identify, which molecules are involved in the disruption of the well-tuned molecular organization at the basis of muscle function in the presence of GAA deficiency, and which are recovered or in the way of recovery after 1 year of therapy. In addition, this study design has permittedto gain a better insight into the muscle dysfunction that characterizes Pompe disease, using the muscle proteome composition of healthy subjects as reference, thus evaluating the effect of ERT in the frame of muscle recovery. The final goal was to identify molecules and pathways that could be targeted to prevent muscle sarcopenia (i.e., the loss of muscle force and mass characterizing muscle disorders and aging) [35].

The lysosomal dysfunction that characterizes LOPD, clearly depicted in Figure 1 and Figure 2 (Electron microscopy), induces profound functional dysregulation in muscle tissue, as highlighted by changes in the protein abundance of 178 out of 900 identified proteins/proteoforms in LOPD patients, 47 of which recovered to normal level after 1 year of ERT therapy. 

Lysosome enlargement in LOPD PRE, hinders the tethering of this organelle to autophagosome, hence blocking the autophagic process and leading to accumulation of undegraded products [10,36]. In skeletal muscle the accumulation of glycogen is not limited to lysosomes, but is also present in autophagic vacuoles containing cytoplasmic degradation products. 

Results from bioinformatic analysis of our proteomic datasets indicate that oxidative phosphorylation is severely impaired both in LOPD PRE and LOPD POST indicating that mitochondria are unable to oxidize substrates for production of ATP that fuels muscle contraction. This occurred before and after treatment and no significant changes on this pathways were observed comparing PRE- and POST-therapy. It is well-known that the electron transport chain capacity per mitochondria declines with age and disease [37,38,39,40]. In our case this decline has to be accounted for Pompe disease since controls were age-matched; thus additional intervention to recover, at least partially, the oxidative phosphorylation, could represent a possible target for this disorder. 

Interestingly, the BAG2 signaling pathway was negatively regulated as well as necroptosis, in LOPD PRE compared to controls, although not markedly. In LOPD POST the former is normalized, and the latter retains its negative regulation. BAG2 signaling involves the inhibition of proteasome activity by heat shock and chaperone protein binding. BAG2 is an inhibitor of the Hsp70-binding E3 ubiquitin ligase CHIP (carboxyl-terminus of Hsp70-interacting protein) regulating protein degradation via the ubiquitin–proteasome pathway [41,42]. Our results suggest that this pathway is inhibited in LOPD PRE and efficiently recovered by the treatment. Another aspect highlighted by this dataset analysis is necroptosis, defined as the passive cell death, that appears inhibited in our patients [43]. Several dysregulated proteins of these pathways were also described in recent studies reviewed by Meena et al. [44,45]. Collectively these data suggest accumulation of undegraded products leading to ER stress and, possibly, to unfolded protein response. 

The activation of unfolded protein response in LOPD PRE, as observed in Figure 6, has been related to increase of HSPA5, VCP, and a number of chaperone molecules involved in the activation of stress response. Activation of unfolded protein response was confirmed by increased levels of CHOP and phosphorylated p38β (Figure 6 and Figure 7B) [46]. Furthermore, based on dysregulation of the muscle proteome, we hypothesized that the increase in giant sarcomeric proteins and HSPG2 could contribute to unfolded protein response and autophagy impairment [47,48,49].

In addition, pathways analysis indicate: sirtuin signaling, RhoA signaling, glycolysis, gluconeogenesis as dysregulated, although not significantly, and PKR/interferon induction, as activated. 

Concerning sirtuin signaling, our results suggest that p38β-FoxO-BNIP3 signaling axis is active in LOPD PRE, but it is also active, although to a lesser extent, in LOPD POST. In cell culture BNIP3 is induced by glucose starvation [50,51]; in our patients, a restriction of the glycolytic pathway was observed after ERT, being the last enzyme of the glycolytic pathway downregulated (PKM). Furthermore, SIRT 1 has a role in regulating oxidative metabolism and in connecting global metabolism to regeneration; and mitochondrial dysfunction is central in regulation of aging responses along several pathways [52]. Thus, these results should be considered as hint for more precise investigation in this direction.

RhoA signaling was significantly dysregulated only in LOPD POST. RhoA is a member of the Rho family of small GTPases, which includes Rho, Rac, and Cdc42 and regulates cytoskeletal dynamics through the action of Rho kinase [53]. PKR/IFN signaling is regulated by RNA-dependent protein kinase (PKR) that is inducible by interferons (IFNs). PKR is a multifunctional serine/threonine kinase which increases in C2C12 myoblast by transactivation of MyoD [54]. The positive signaling could be associated to previous observations concerning numbers of satellite cells in skeletal muscle biopsies in LOPD patients indicating retention of the satellite cell niche, thus making muscle regeneration possible [55]. Moreover, in LOPD POST vs. LOPD PRE glycolysis and gluconeogenesis were inhibited suggesting that the treatment specifically targets these pathways. 

The signaling pathway analysis indicates molecules involved in events, but results are not associated with a specific disorder. The relationship between protein at variance and disorders is provided by the disease and biofunction analysis of proteomic datasets. As indicated in results, among a huge number of disorders displayed in Appendix A, the top 3 were sliding of myofilaments, hereditary myopathy, and muscle contraction. In all of them proteins involved were negatively regulated. Particularly sliding of filaments appeared strongly inhibited. Despite the positive increment of cytoskeletal proteins and troponin located in fast fibers, molecules involved in the antiparallel organization of the Z-Disk and controlling the assembly and alignment of the sarcomere during contraction, were decreased, suggesting that, in this time frame, the recovery of muscle function was not completely achieved. Furthermore, the different abundance of contractile and sarcomeric proteins, revealed by proteome analysis (increment of MYL6B, ACTN3, OBSCN and DMD, and decrease of MYL1 and TNNT3 in LOPD POST vs. LOPD PRE), suggests a rearrangement of fiber type distribution (see Appendix A), indicating that muscle tissue counteracts negative signals that promote fiber type rearrangement, in an attempt to recover or maintain its mass and function [56]. However, channel alterations were still present, as indicated by increased RYR1 and CASQ1, thus worsening of Ca^2+^ balance would induce disruption of the triad junction, as shown by electron microscopy. Furthermore, the increment of specific proteins localized in M bands indicates that the ordered structure of striated muscle is not yet achieved and that the fast and unidirectional development of force and motion during skeletal muscle contraction is hampered, resulting in muscle weakness [57,58,59]. However, after 1 year of therapy the fine tuning of proteostasis (i.e, the balance between protein synthesis and degradation), crucial for muscle function and regeneration was not achieved, suggesting that the muscle requires longer treatment and additional supports to completely recover or at least maintain its function [60]. 

ERT treatment improves HSPG2 levels, recently associated with lysosomal storage disorders and involved in the regulation of autophagy mediated by mTOR signaling [47,48,49]. 

The autophagic signaling is committed to the autophagosome formation but to be finalized the contribution of the lysosome is crucial [61]. In our patients, the membrane remodeling was active due to high levels of LC3BII/LC3BI. The accumulation of LC3II is consistent with accumulation of autophagosomes, however p62 also accumulated, thus indicating that it was not consumed in the autophagic process. Regarding lysosomes, LAMP2 was overexpressed, both in LOPD PRE and LOPD POST, suggesting that the lysosome biogenesis was active. The discrepancy was observed in molecules involved in lysosome tethering (VAMP8, SNAP29, STX17 and GORASP2) and BNIP3. It has been demonstrated that BNIP3 may play an important role in the regulation of autophagosome-lysosome fusion [62,63]. BNIP3, both monomer and dimer, was surprisingly overabundant in LOPD PRE. In cell models it has been recently described that BNIP3 overexpression was able to block the autophagosome-lysosome fusion through inhibition of the interaction between SNAP29 and VAMP8, resulting in the accumulation of autophagosomes [36]. In our patients, monomeric and dimeric BNIP3 were increased and it appears that in response to energy stress, p38β phosphorylates FoxO3, in LOPD PRE, partially retained in LOPD POST [64]. Our results suggest that p38β-FoxO-BNIP3 signaling axis is active in LOPD PRE, but it is also active, although to a lesser extent, in LOPD POST. In cell cultures BNIP3 is induced by glucose starvation [50,51]; in our patients, a restriction of the glycolytic pathway was observed, being the last enzyme of the glycolytic pathway downregulated (PKM). These results should be considered as hint for more precise investigation in this direction.

## 4. Materials and Methods

### 4.1. Ethics Statement 

The study protocol and consent forms were approved by the ethics committees of the Fondazione IRCCS Istituto Neurologico Carlo Besta (protocol code N. 32, 7 September 2016) and the Fondazione IRCCS Ca’ Granda, Ospedale Maggiore Policlinico, Milan (protocol code N. 632, 13 March 2012). Informed consent for muscle biopsy and biopsy storage for research was obtained in all cases from patients. The investigation was conducted according to the Declaration of Helsinki.

### 4.2. Patients 

Skeletal muscle biopsies from 10 LOPD patients were collected by needle biopsy, frozen in pre-chilled isopentane, and stored in liquid nitrogen. All patients underwent two muscle biopsies according to the following criteria: the first biopsy was performed for diagnostic purposes before the study was designed, namely 2 months – 3 years before starting ERT; the second one year after ERT administration and 7–8 days after the latest ERT infusion. Pompe Disease diagnosis was confirmed at molecular level in all patients. ERT was administered by intravenous infusions of alpha-glucosidase (Myozyme^®^; Sanofi Genzyme, Cambridge, MA, USA) at a dose of 20 mg/kg every other week (Appendix A).

Biopsies from 15 adult healthy males subjects were taken in absence of strenuous exercise, samples were frozen in liquid nitrogen (Appendix A).

### 4.3. Electron Microscopy 

Biopsy material derived from quadriceps muscle of 3 patients was fixed in 2.5% glutaraldehyde in 0.1 M sodium phosphate buffer (pH 7.3) and post-fixed with 2% osmium tetroxide in the same buffer. The specimens were washed in distilled water, dehydrated through an ascending series of ethanol, embedded in Epon-Araldite resin (Zeiss, Oberkochen, Germany) and oriented for longitudinal sectioning. Ultrastructural analysis was carried out on ultrathin sections after staining with uranyl acetate and lead citrate under the transmission electron microscope Zeiss EM 10 (Zeiss, Oberkochen, Germany).

### 4.4. Proteomic Analysis

Soluble extracts from frozen muscle were obtained from patients with late-onset Pompe disease; in particular, 10 LOPD patients before (LOPD PRE) and 10 after enzymatic treatment (LOPD POST) and from 15 controls (CTR). Sample extracts were analyzed by 2D-DIGE and label-free LC–MS/MS to evaluate proteome changes in pre- and post-ERT patients as compared to age-matched healthy controls.

#### 4.4.1. Protein Extraction

After tissue homogenization, for 2D-DIGE, each sample from each subject was suspended in lysis buffer (7 M urea, 2 M thiourea, 4% 3-[(3-cholamidopropyl)dimethylammonio]-1-propanesulfonate (CHAPS), 30 mM Tris, and 1 mM PMSF) and solubilized by sonication on ice. Proteins were selectively precipitated using PlusOne 2D-Clean up Kit (GE Healthcare, Little Chalfont, UK), in order to remove non-protein impurities, and resuspended in lysis buffer. The pH of the protein extracts was adjusted to pH 8.5 by addition of 1 M NaOH. Protein concentration was determined by PlusOne 2D-Quant Kit (GE Healthcare, Little Chalfont, UK). 

#### 4.4.2. Two-Dimensional Difference in Gel Electrophoresis

Soluble extracts from each frozen muscle were analyzed by quantitative 2D-DIGE, followed by mass spectrometry. Protein minimal labeling with cyanine dyes (Cy3 and Cy5), 2D separation and analyses were performed, as described previously [26]. Briefly, the proteins extracted (50 µg) from each individual sample were labeled with Cy5, while internal standards were generated by pooling (50 µg) individual samples that were Cy3-labeled. Samples were separated on 24 cm, 3–10 nonlinear immobilized pH gradient (IPG) strips; each individual sample was run in triplicate to minimize the inter-gel variability and increase results reliability. Image analysis was performed using DeCyder 6.5 software (GE Healthcare, Little Chalfont, UK). All gel images were imported into individual differential in-gel analysis (DIA) workspaces. Using the Batch Processor tool, automated detection of protein spots was performed with the following filter settings: estimated number of spots: 10,000, exclusion slope > 1.2; minimal area cutoff < 200 and peak height: 100,000. DIA workspaces were then manually edited to eliminate gel artifacts (e.g., plate scratches and dust specks) and include any incorrectly excluded spot. The resulting spot maps (containing the spot identifiers, locations, and normalized volumes for all protein spots in each channel of each gel) were further processed in the biological variation analysis (BVA) module. Individual DIA workspaces for all analytical gels were imported into the BVA module. The BVA workspace was used for inter-gel protein spot matching. Statistical analysis was performed using the DeCyder 1.0 (GE Healthcare, Little Chalfont, UK) extended data analysis (EDA) module. Protein filters were set to select only those protein spots that matched 90% of the gel images and these protein spots were included in data analysis. Statistically significant differences were computed by analysis of variance (ANOVA) and Tukey’s tests (*p* < 0.05) for the comparison between: CTR vs. LOPD PRE, CTR vs. LOPD POST and LOPD PRE vs. LOPD POST. In case the ANOVA test was not applicable, the non-parametric Kruskal–Wallis test was used. False discovery rate (FDR) was applied to correct for multiple tests to reduce the overall error. Statistically changed proteins underwent the power analysis, and only spots reaching a sensitivity cut-off >0.8 were considered as differentially expressed. 

Proteins of interest were identified by matrix-assisted laser desorption/ionization–time-of-flight (MALDI-ToF)/MS. For protein identification, semi preparative gels were loaded with unlabeled sample (400 µg per strip); electrophoretic conditions were the same of 2D-DIGE. Gels were stained with a total-protein fluorescent stain, KryptonTM (Thermo Fisher Scientific, Waltham, MA, USA), and images were acquired using a Typhoon 9200 laser scanner. Spots of interest were excised from gel using the Ettan spot picker robotic system (GE Healthcare, Little Chalfont, UK), destained in 50% methanol/50 mM ammonium bicarbonate (AMBIC), and incubated with 30 µL of 6 ng/mL trypsin (Promega, Fitchburg, MA, USA) dissolved in 10 mM AMBIC for 16 h at 37 °C. Released peptides were subjected to reverse phase chromatography (Zip-Tip C18 micro; Millipore, Burlington, MA, USA), eluted with 50% acetonitrile (ACN)/0.1% trifluoroacetic acid. Peptide mixture (1 µL) was diluted in an equal volume of 10 mg/mL alpha-cyano-4-hydroxycinnamic acid matrix dissolved in 70% ACN/30% citric acid and processed on a Ultraflex III MALDI-ToF/ToF (Bruker Daltonics, Billerica, MA, USA) mass spectrometer. MS was performed at an accelerating voltage of 20 kV and spectra were externally calibrated using Peptide Mix calibration mixture (Bruker Daltonics, Billerica, MA, USA); 1000 laser shots were taken per spectrum. Spectra were processed by a FlexAnalysis software v. 3.0 (Bruker Daltonics, Billerica, MA, USA) setting the signal to noise threshold value to 6 and search was carried out by correlation of uninterpreted spectra to Homo sapiens entries in NCBIprot database 20180429 (152.462.470 sequences; 55.858.910.152 residues) using BioTools v. 3.2 (Bruker Daltonics, Billerica, MA, USA) interfaced to the on-line MASCOT software, which utilizes a robust probabilistic scoring algorithm. The significance threshold was set at a *p*-value < 0.05. No mass and pI (isoelectric point) constraints were applied and trypsin was set as enzyme. One missed cleavage per peptide was allowed and carbamidomethylation was set as fixed modification while methionine oxidation as variable modification. Mass tolerance was set at 30 ppm for MS spectra (For further information about peptide mass fingerprinting data are listed in Appendix A). 

#### 4.4.3. Label-Free Liquid Chromatography with Tandem Mass Spectrometry

Proteins were precipitated with PlusOne 2D-Clean up kit (GE Healthcare, Little Chalfont, UK), resuspended in 50 mM ammonium bicarbonate and 0.1% Rapigest SF surfactant (Waters Corporetion, Milford, MA, USA). Dithiotreitol (DTT) was added to a final concentration of 5 mM for cysteine reduction, and samples were incubated for 45 min at 60 °C. Iodoacetamide (IAA) was added to a final concentration of 15 mM and incubated for 45 min in the dark. Proteins were digested with sequence grade trypsin (Promega, Fitchburg, WI, USA) for 16 h at 37 °C using a protein:trypsin ratio of 50:1. RapiGest was precipitated by adding trifluoroacetic acid (TFA) to a final concentration of 0.5% and samples incubated for 45 min at 37 °C. After centrifugation at 13,000 rpm for 10 min, the supernatants were recovered, and the peptide concentration was determined by Pierce™ Quantitative Colorimetric Peptide Assay (Thermo Fisher Scientific, Waltham, MA, USA). 1 µg per sample was utilized for MS analysis. LC-ESI-MS/MS analysis was performed on a Dionex UltiMate 3000 HPLC System with an Easy Spray PepMap RSLC C18 column (250 mm, internal diameter of 75 μm) (Thermo Fisher Scientific, Waltham, MA, USA). Gradient: 5% ACN in 0.1% formic acid for 10 min, 5–35% ACN in 0.1% formic acid for 139 min, 35–60% ACN in 0.1% formic for 40 min, 60–100% ACN for 1 min, 100% ACN for 10 min at a flow rate of 0.3 μL/min. The eluate was electrosprayed into an Orbitrap Tribrid Fusion through a nanoelectrospray ion source (Thermo Fisher Scientific, Waltham, MA, USA). The LTQ-Orbitrap operated in positive mode, in data-dependent acquisition mode to automatically alternate between a full scan (350–2000 m/z) in the Orbitrap (at resolution 60,000, AGC target 1,000,000) and subsequent collision-induced dissociation (CID) MS/MS in the linear ion trap of the 20 most intense peaks from full scan (normalized collision energy of 35%, 10 ms activation). Isolation window: 3 Da, unassigned charge states: rejected, charge state 1: rejected, charge states 2+, 3+, 4+: not rejected; dynamic exclusion enabled (60 s, exclusion list size: 200). Three technical replicates for each sample were acquired. Mass spectra were analyzed using MaxQuant software (v. 1.6.3.3, Max Planck Institute of Biochemistry, Martinsried, Germany) [65]. The initial maximum allowed mass deviation was set to 6 ppm for monoisotopic precursor ions and 0.5 Da for MS/MS peaks. Enzyme specificity was set to trypsin/*p*, and a maximum of two missed cleavages were allowed. Carbamidomethylation was set as a fixed modification, while N-terminal acetylation and methionine oxidation were set as variable modifications. The spectra were searched by the Andromeda search engine against the Homo Sapiens Uniprot sequence database (release 22.10.2018, https://www.uniprot.org/news/2018/10/10/release, accessed on 8 February 2021). Protein identification required at least one unique or razor peptide per protein group. Quantification in MaxQuant was performed using the built in extracted ion chromatogram (XIC)-based label free quantification (LFQ) algorithm using fast LFQ [66]. The required FDR was set to 1% at the peptide, 1% at the protein and 1% at the site-modification level, and the minimum required peptide length was set to 7 amino acids. Statistical analyses were performed using the Perseus software (v. 1.4.0.6, Max Planck Institute of Biochemistry, Martinsried, Germany) [67]. Only proteins present and quantified in at least two out of three technical repeats were considered as positively identified in a sample. For each experimental group, the proteins identified in at least 80% of samples were considered. Statistically significant differences were computed by ANOVA and FDR (*p* < 0.05) followed by Tukey’s post-hoc test (*p* < 0.05) (Appendix A).

#### 4.4.4. Ingenuity Pathway Analysis

Functional and network analyses of statistically significant protein expression changes were performed through Ingenuity Pathway Analysis (IPA) software (Qiagen, Redwood City, CA, USA). In brief, data sets with protein identifiers, statistical test *p*-values and fold change values calculated from the mass-spectrometry experiment were analyzed by IPA. The “core analysis” function was used to interpret the data, through the analysis of biological processes, canonical pathways, upstream transcriptional regulators enriched with differentially regulated proteins. The “comparison analysis” function was used to visualize and identify significant proteins or regulators across experimental conditions. *p*-values were calculated using a right-tailed Fisher’s exact test. Activation z-score was used to predict the activation/inhibition of a pathway/function/regulator [68]. A Fisher’s exact test *p*-value < 0.05 and a z-score <−2 and >2, which takes into account the directionality of the effect observed, were considered statistically significant.

### 4.5. Immunoblotting

Protein extracts (50 μg) from pooled LOPD PRE, LOPD POST, and CTR were loaded in triplicate and resolved on 8%, 12%, 12–18% and 10-16% gradient gels of sodium dodecyl sulfate-polyacrylamide gel electrophoresis (SDS-PAGE), according to protein molecular weight. The resolved proteins were then transferred to a 0.45 μm polyvinylidene difluoride (PVDF, Merck Millipore, Burlington, MA, USA.) membrane. The membrane was blocked with 5% bovine serum albumin for 1 h at room temperature and incubated, overnight at 4 °C, with rabbit, mouse, or goat polyclonal primary antibodies (Cell Signaling Technology, Danvers, MA, USA, except when differently stated) as follows: anti-microtubule associated protein 1 light chain 3 beta (LC3B/MAP1LC3B, 2775), 1:1000; anti-ubiquitin-binding protein p62/sequestosome 1 (p62/SQSTM1, P0067; Sigma–Aldrich, St. Louis, MO, USA), 1:1000; anti-sintaxin 17 (STX17, HPA001204; Sigma–Aldrich, St. Louis, MO, USA), 1:1000; anti-synaptosomal-associated protein 29 kDa (SNAP29, sc-135564; Santa Cruz Biotechnology, Dallas, TX, USA), 1:500; anti-endobrevin (VAMP8, sc-166820; Santa Cruz Biotechnology, Dallas, TX, USA), 1:500; anti-golgi reassembly-stacking protein 2 (GORASP2, HPA035275; Sigma–Aldrich, St. Louis, MO, USA), 1:1000; anti-lysosome-associated membrane protein 2 (LAMP2, 49067), 1:1000; anti-BCL2/adenovirus E1B 19 kDa protein-interacting protein 3 (BNIP3, B7931; Sigma–Aldrich, St. Louis, MO, USA), 1:1000; anti-protein kinase AMP-activated catalytic subunit alpha 1 (AMPK/PRKAA1, 2532), 1:1000; anti-phosphoAMPK (2531), 1:1000; anti-p38 mitogen-activated protein kinases 11 (p38β/MAPK11, 8690), 1:1000; anti-phosphop38β (9215), 1:1000; anti-forkhead box O3 (FoxO3, 2497), 1:1000; anti-phosphoFoxO3 (9464), 1:1000; anti-multi transcriptional factor C/EBP-homologous protein (CHOP/DDIT3, 2895), 1:1000; anti-GLUL (6640; Santa Cruz Biotechnology, Dallas, TX, USA), 1:500. After washing, membranes were incubated with anti-rabbit or anti-mouse (GE Healthcare, Little Chalfont, UK) or anti-goat (Santa Cruz Biotechnology, Dallas, TX, USA) secondary antibodies, conjugated with horseradish peroxidase. Signals were visualized by chemiluminescence using the ECL Prime detection kit and the ImageQuant LAS 4000 mini (GE Healthcare, Little Chalfont, UK) digital imaging system. Band quantification was performed using the Image Quant TL (Molecular Dynamics, Ragusa, Italy) software followed by statistical analysis (ANOVA + Tukey, *n* = 3, *p* < 0.05). Band intensities were normalized against the total amount of proteins stained by Sypro Ruby Blot Stain (Life Technologies Europe BV, Monza, Italy).

## 5. Conclusions

The cascade of effects described above in LOPD PRE starts with a very simple problem: GAA deficiency. Our results indicate that after 1 year of ERT, besides removing this primary cause, targeting secondary effects could improve patient’s outcome. The present study also suggests that prolonged therapy is required to restore muscle function. In addition, commencing the therapy at earlier stages of the disease is likely to help improve its long-term outcome. Our patients were in fact all adults with disease duration of several years. 

Furthermore, our results confirm recent developments in Pompe disease treatment that suggest a combination of therapies which, by targeting metabolism, autophagy and lysosomal tethering, can contribute to muscle proteostasis maintenance, counteract muscle sarcopenia, and improve outcome of the necessary GAA replacement. 

However, further studies addressing the role of post translation modifications and epigenetic changes will be required to more precisely define changes in the context of response to the hGAA replacement.

We are aware of the limitations of our study, which involves a restricted number of patients from whom small biopsies were available. Despite this, our results provide a better understanding of dysregulations occurring in the muscle tissue associated with disrupted muscle integrity and the consequent loss of function in LOPD only partially recovered by 1 year of ERT. 

## Figures and Tables

**Figure 1 ijms-22-02850-f001:**
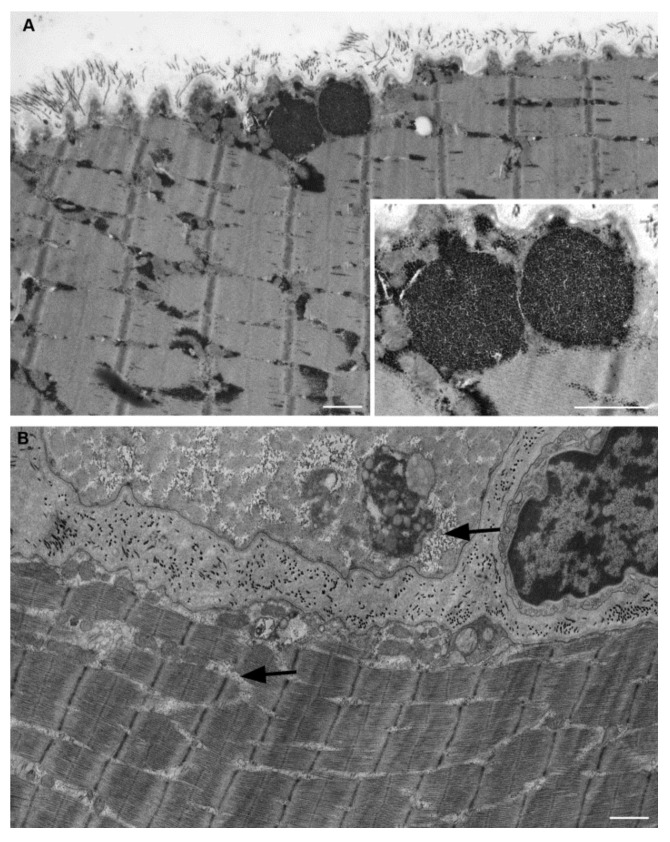
Electron microscopy of muscle fibers in longitudinal section of a patient with Pompe disease. (**A**) Pre-treatment muscle fiber. Two glycogen-filled lysosomes under sarcolemma are enlarged in the inset. Massive glycogen accumulations are also present within intermyofibrillar spaces. (**B**) Post treatment muscle fiber shows relevant glycogen reduction. A low amount of glycogen granules (arrows) still appears located within some intermyofibrillar spaces. Scale bar = 1 µm. Inset scale bar = 0.5 µm.

**Figure 2 ijms-22-02850-f002:**
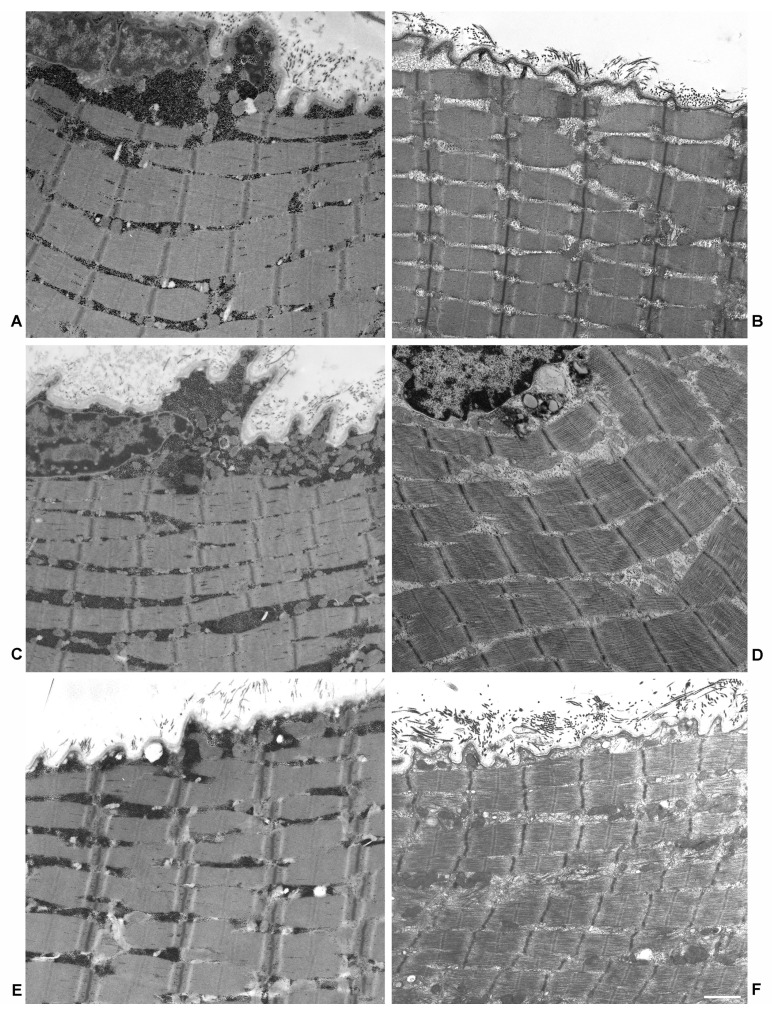
Electron microscopy of muscle fibers in longitudinal section of three Pompe disease patients. (**A**,**C**,**E**) are from pre-treatment patients. Massive glycogen accumulations are evident both in the cytoplasm immediately beneath the plasma membrane and in the intermyofibrillar spaces. (**B**,**D**,**F**) Micrographs from the same patients after treatment show a relevant glycogen reduction. Bar = 1.5 µm.

**Figure 3 ijms-22-02850-f003:**
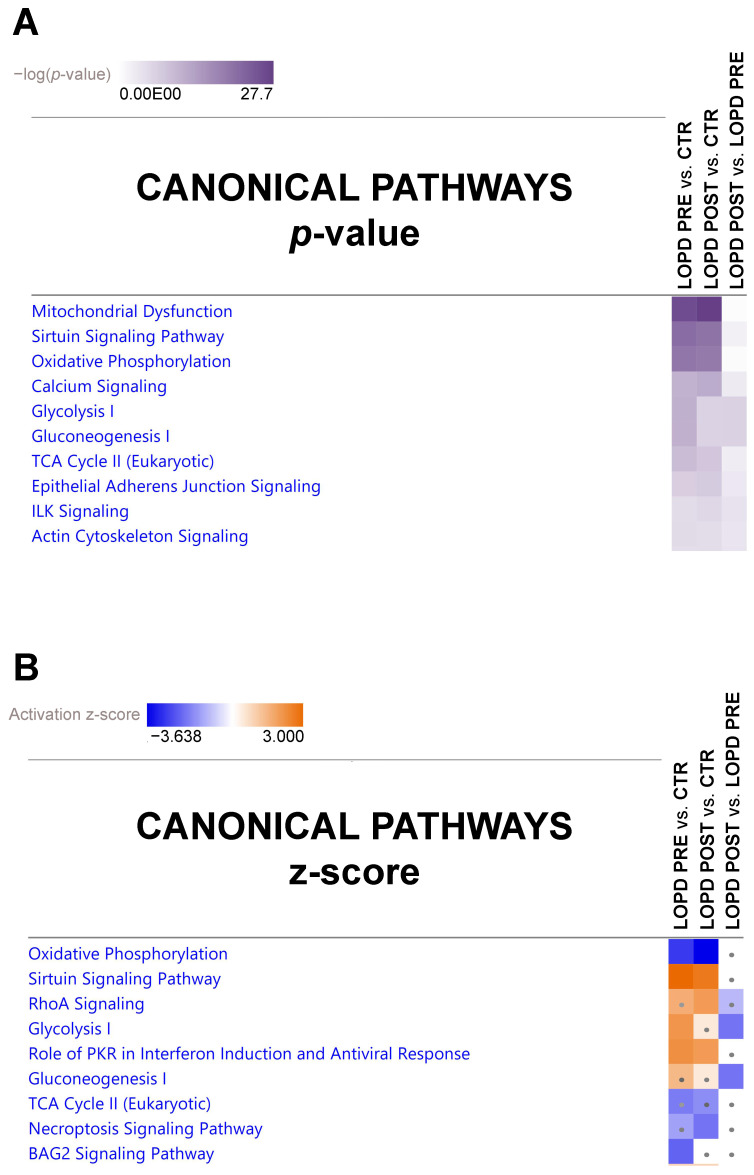
Canonical pathway heatmap displaying the most significant results (*p*-value in panel (**A**) and z-scores in panel (**B**)) across datasets in LOPD PRE vs. CTR, LOPD POST vs. CTR. and LOPD POST vs. LOPD PRE. The orange and blue colored rectangles indicate predicted pathway activation or predicted inhibition, respectively, via the z-score statistic (z-score ≥ 2, ≤ −2). Dots are non-significant values for z-score.

**Figure 4 ijms-22-02850-f004:**
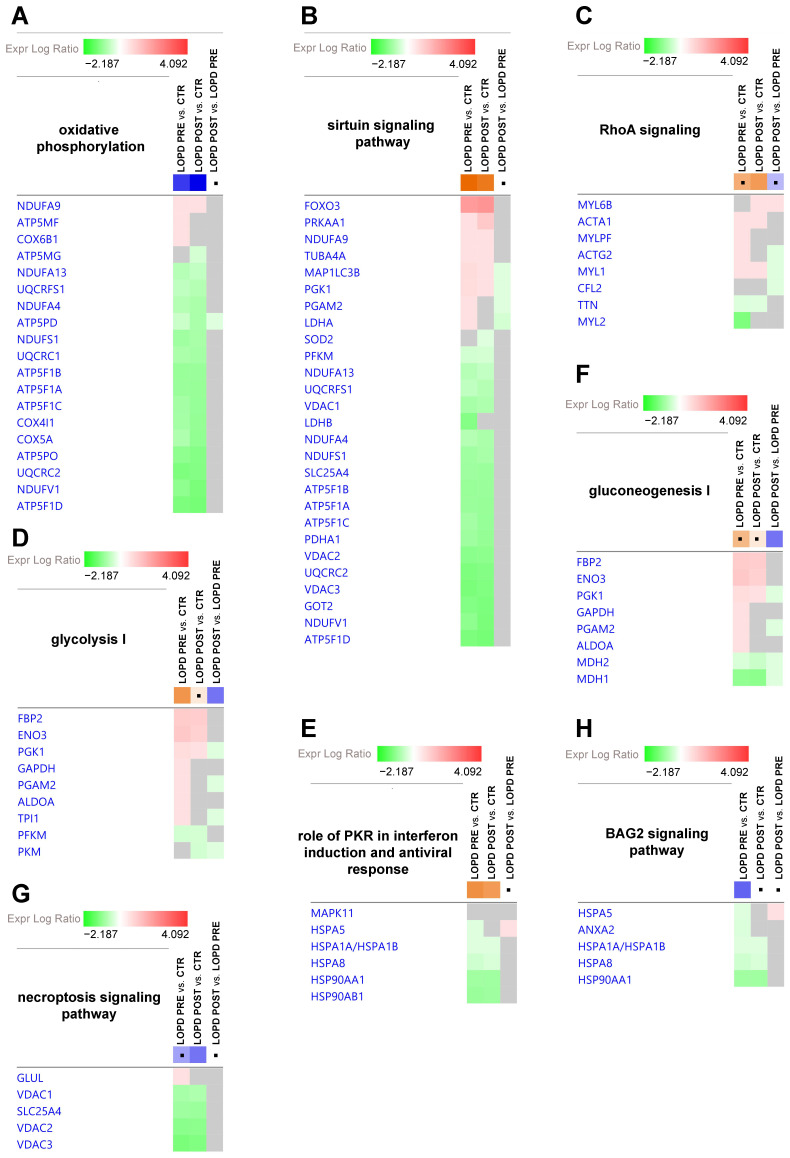
Gene heatmap enables to visualize protein expression data in a specific canonical pathway, across multiple analyses at one time. (**A**) Oxidative phosphorylation. (**B**) Sirtuin signaling pathway. (**C**) RhoA signaling. (**D**) Glycolysis I. (**E**) Role of PKR in interferon induction and antiviral response. (**F**) Gluconeogenesis I. (**G**) Necroptosis signaling pathway. (**H**) BAG2 signaling pathway. Green and red colors refer respectively to decrease or increase for each individual protein statistically changed in our proteomics dataset. Grey color indicates changes that are statistically not significant for *p*-value (ANOVA followed by Tukey’s post-hoc test (*p* < 0.05)). Dots indicate non-significant values for z-score (z-score ≥ 2, ≤ −2).

**Figure 5 ijms-22-02850-f005:**
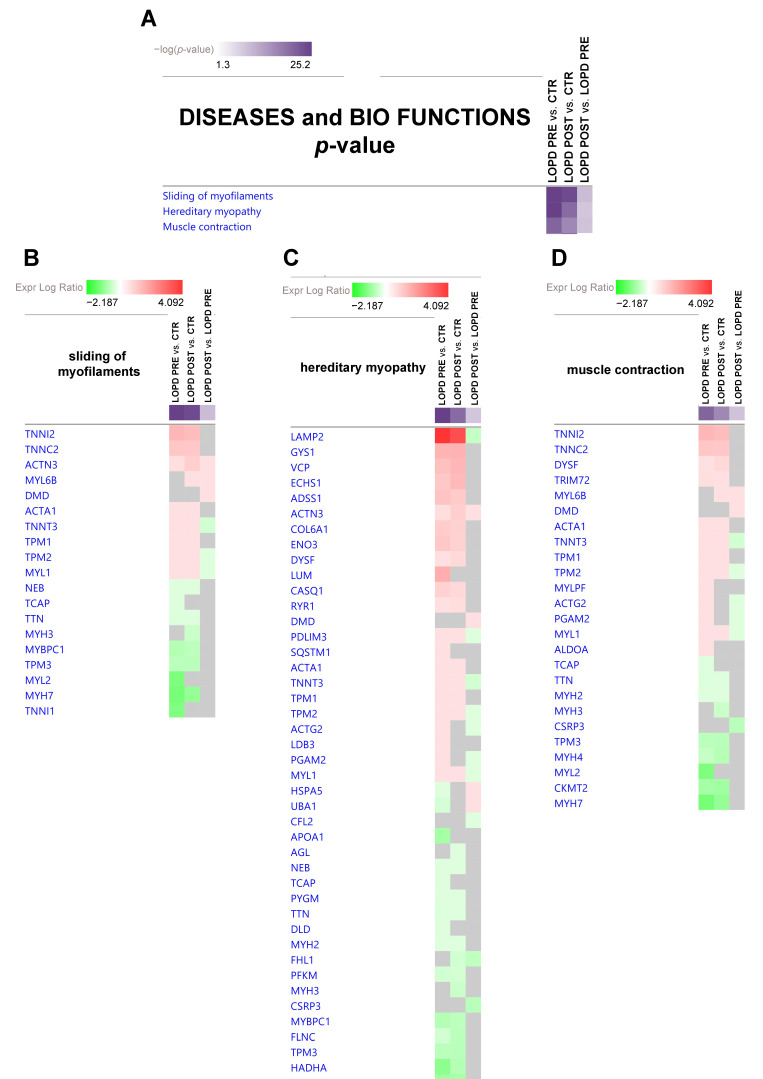
(**A**) Diseases and bio functions heatmap displaying the most significant results (*p*-value, Appendix A) across datasets in LOPD PRE vs. CTR, LOPD POST vs. CTR, and LOPD POST vs. LOPD PRE. (**B**,**C**,**D**) Gene heatmap enables to visualize protein expression data in a specific disease and biofunction, across multiple analyses at one time. Green and red colors refer to decrease or increase for each individual protein statistically changed in our proteomics dataset. Grey color indicates changes statistically non-significant for *p*-value (ANOVA followed by Tukey’s post-hoc test (*p* < 0.05)). Dots are non-significant values for z-score (z-score ≥ 2, ≤ −2).

**Figure 6 ijms-22-02850-f006:**
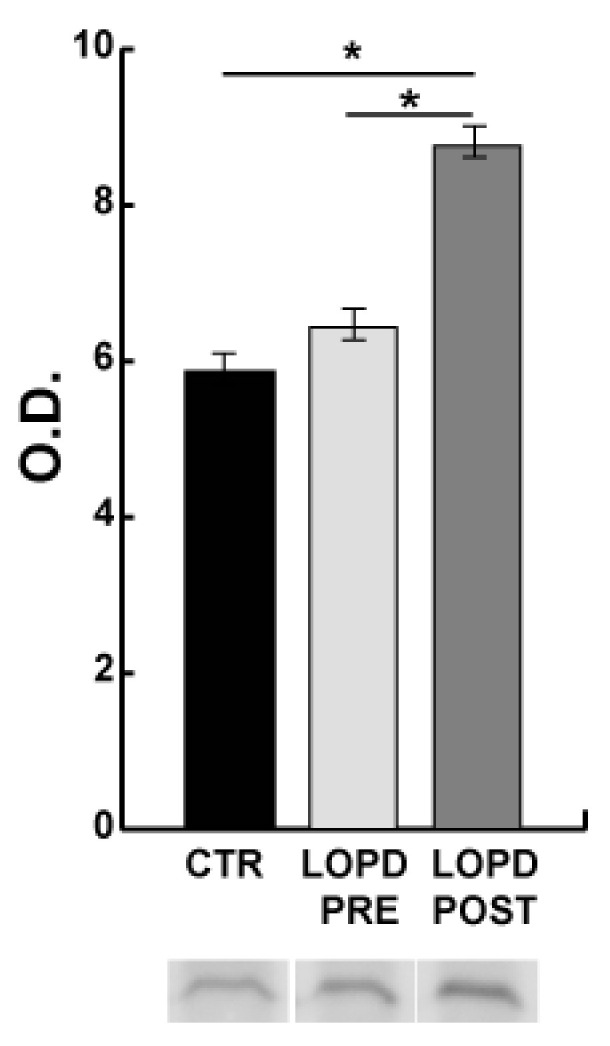
Endoplasmic reticulum (ER) stress. Representative immunoblot images and histograms (mean ± SD) showing protein abundance of C/EBP-homologous protein (CHOP/DDIT3). The data were normalized against the total amount of loaded proteins stained with Sypro Ruby. Statistical analysis was performed by ANOVA and Tukey’s test (* *p* < 0.05). To improve visualization blot images have been cropped, full-length images are available in Appendix A. O.D. = optical density.

**Figure 7 ijms-22-02850-f007:**
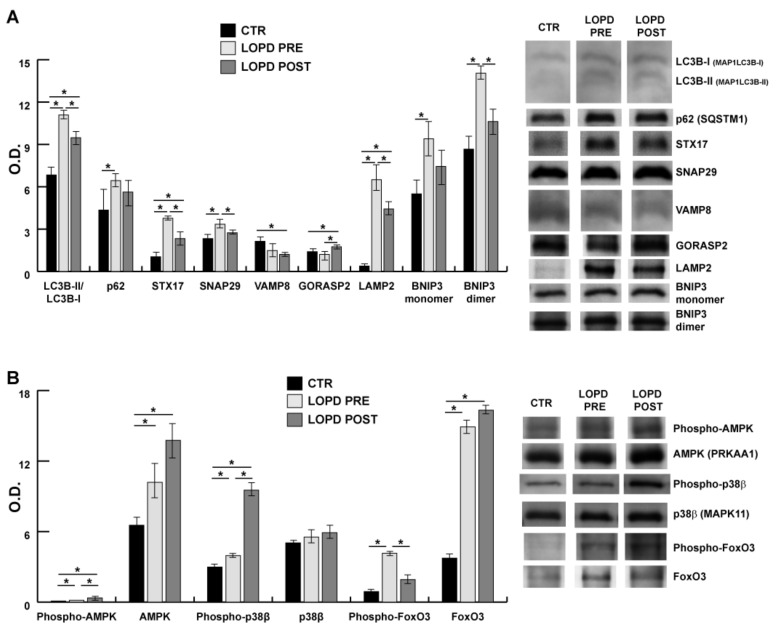
Lysosomal autophagy and apoptosis. Representative immunoblot images (cropped images) and histograms (mean ± SD) showing protein abundance of: (**A**) microtubule associated protein 1 light chain 3 beta (LC3B/MAP1LC3B), ubiquitin-binding protein p62/sequestosome 1 (p62/SQSTM1), sintaxin 17 (STX17), synaptosomal-associated protein 29 kDa (SNAP29), endobrevin (VAMP8), Golgi reassembly-stacking protein 2 (GORASP2), lysosome-associated membrane protein 2 (LAMP2), BCL2/adenovirus E1B 19 kDa protein-interacting protein 3 (BNIP3) in CTR, pre- and post-ERT LOPD; and (**B**) protein kinase AMP-activated catalytic subunit alpha 1 (AMPK/PRKKA1), p38 mitogen-activated protein kinases 11 (p38β/MAPK11), and fork head box O3 (FoxO3) in CTR, LOPD PRE and LOPD POST. The data were normalized against the total amount of loaded proteins stained with Sypro Ruby. Statistical analysis was performed by ANOVA and Tukey’s test (* *p* < 0.05). Blot images have been cropped for a better visualization, full-length images are available in Appendix A. O.D. = optical density.

## Data Availability

The data presented in this study are available in Appendix A. Study data from this human study other than those published in this work are under privacy regulations but can be obtained on a case-to case basis upon reasonable request from the corresponding author. See also “MDPI Research Data Policies” at https://www.mdpi.com/ethics (accessed on 8 February 2021).

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
