# Peer review of "Muscle Proteomic Profile before and after Enzyme Replacement Therapy in Late-Onset Pompe Disease"

_ijms, 2021, doi:10.3390/ijms22062850_

Round 1
Reviewer 1 Report
I think the authors should mentioned that effect of duration of the disease and the time of ERT commence and whether they expect protein dysregulation to change. If dysregulation is less depending on when ERT is started, it would further support institution of Pompe screening as NBS.
Author Response
The authors thank the Reviewer for his/her constructive comments. Point-by-point responses to the reviewer’s comments are provided below, as requested.
Reviewer 1.
Comments and Suggestions for Authors
I think the authors should mentioned that effect of duration of the disease and the time of ERT commence and whether they expect protein dysregulation to change. If dysregulation is less depending on when ERT is started, it would further support institution of Pompe screening as NBS.
Author response
We would like to thank the Reviewer for his/her comment that we have taken into consideration. The time when ERT starts is important since, physiologically, muscle declines with age and this process can be accelerated by GAA deficiency. NBS Pompe screening can promote the early treatment of patients and preserve muscle from a severe proteostasis dysregulation contributing to sarcopenia. We have now introduced this concept into the conclusions.
Reviewer 2 Report
This article is based on highly comprehensive investigations of the proteasome profile in skeletal muscle of patients with late-onset Pompe disease (LOPD). Investigations were carried out prior to the initiation of enzyme replacement therapy (ERT) and following a one-year ERT administration. Protein expression in muscle tissue was assessed in the same patient cohort using an identical set of methods. Proteasome data before and after ERT were compared to healthy control subjects.
I strongly commend the authors for their meticulous approach and methodological soundness which are both clearly reflected by this manuscript. Obviously, the superior aim of the study was to identify disease-related intracellular alterations of the proteasome that further elucidate the molecular pathology of LOPD. Without doubt, it is an intriguing approach to also investigate the effect of ERT on these changes. Thus, protein clusters or functional pathways may be identified which differentially respond to enzyme supplementation, possibly revealing preliminary information that helps to understand why clinical effectiveness of current ERT appears to decline after several years of treatment.
Although the manuscript provides abundant new data and clearly has the potential to substantially contribute to the field there are several major issues and minor flaws that have to be addressed. Most of them relate to readability, clarity of data presentation, language, and - most important - data interpretation with regard to both pathophysiological and clinical significance.
The following remarks are structured according to the main chapters of the manuscript:
- Introduction:
- The overall aim of the study (as I tried to summarise it further above) is not clearly depicted in the introduction. In particular, it is not yet properly addressed that this study's results might help to understand why clinical effectiveness of ERT decreases after several months, resulting in a "plateau phase" which may last for several years before progressive decline may reoccur in a subset of patients.
- It should be made more clear that the study aimed to specifically and directly compare pre-ERT and post-ERT characteristics of the muscle proteasome.
- I am quite unhappy with the term "engulfment of lysosomes" which is used several times in a way which I think is incorrect. Engulfment is what what happens to any material intracellularly taken up by lysosomes, but this term does not properly describe what happens to the lysosomes themselves in Pompe disease or other autophagic processes, respectively. Thus, terms like "degradation", "dysfunction" or "involvement" (of lysosomes may be more appropriate).
- Line 83: In LOPD, there is no "unavailability of glycogen" since degradation of cytoplasmatic glycogen is not affected by the mutation.
- Materials and methods:
- Line 542: "one" and "1" are redundant.
- Line 543: The abbreviation "PD" for Pompe disease should be avoided since it usually stands for Parkinson's disease.
- Line 586: The abbreviation CTR has not been introduced before, please amend.
- Line 615: The abbreviation "pI" has not yet been introduced.
- Line 638: Since both devices are manufactured by Thermo Fisher it is sufficient to mention the company once.
- Line 665: There is a dot between 12% and 12-18% but it should be a comma.
- Results:
- Unfortunately, the results section is almost impenetrable. It is lacking a clear-cut structure and the vast number of abbreviations is of no help, too. I strongly recommend to thoroughly revise this section in order to improve readability. Results should be presented in a way that makes it easier to understand the facts and to grip the principles that (may) lie behind these facts. Otherwise, it is impossible to get an idea which of the observations might be meaningful (and which might not). Graphical presentation of results as it is provided in Tables 3 and 4 (with horizontal bars indicating any change from LOPD PRE to LOPD POST) is much more convenient than data presentation in Figure 2 and Table 1. It is prudent to enable the reader to get a visual impression of those changes that reach statistical significance on group comparison (LOPD PRE vs. CTR, LOPD PRE vs. LOPD POST). Regarding direct comparison between LOPD POST and CTR the respective data don't have to be presented in detail as long as they show no difference from the LOPD PRE-CTR analysis. What the reader ist interest in is any (significant) change that is likely to be caused by ERT. For graphical presentation of data, distinct proteins should be categorised as it has been already done (in subchapters 2.1.1-2.1.4) but also grouped according to the specific difference compared to CTR or LOPD PRE vs. POST. If this is carried appropriately, the large number of abbreviations may be reduced for the sake of readability.
- Figure 2 (if it is kept): glycolytic (spelling error).
- Table 1 (if it is kept): Please write "%" out in the heading, i. e. "Percentage".
- Line 101: Please replace "have been" by "were".
- Line 106: Please replace "over" by "out of".
- Line 107: Please replace "changed" by "altered".
- Line 320: microscopy.
- Line 368: Please stick to LOPD PRE and LOPD POST and do not use PRE- and POST-LOPD.
- Discussion:
- The discussion section still lacks a red thread that goes from the initial hypotheses to a concise summary of findings and to data interpretation regarding pathophysiology and clinical significance. The authors should outline which portions of the proteasome show difference as compared to CTR, which subgroups of proteins obviously "respond" to ERT and how this might translate into a), functional improvement which is partial at least, and b), persisting cellular dysfunction on the proteasome level that possibly explains why clinical effectiveness of ERT is still suboptimal or limited.
- Line 425: "FHL3" is not helpful here, please delete.
- Line 428: I wonder why the authors specifically relate cytoskeletal proteins and troponin to fast fibres here.
- Line 431: "(...) by comparing the different abundance of contractile and sarcomeric proteins in the proteome of the same patient before and after ERT (...)". Same patients, I suppose?
- Line 432: What does DMD stand for?
- Line 436: "It is well known that patients are characterized by muscle impairment which is initially ameliorated by ERT but not maintained after one year of treatment [41]." This statement is incorrect since extensive data support the clinical observation of a plateau phase in which motor function is maintained for more than a year.
- Line 450: "As pointed out by proteomic analyses, the major drawback is represented by metabolism." I guess the authors want to convey that proteomic analyses indicate cell metabolism to be primarily disrupted by LOPD as reflected by the plethora of "metabolic" proteins which show altered expression. Please put this more clear-cut.
- Line 473: "Concerning stress proteins, a number of proteins involved in proper folding are severely dysregulated and not recovered after ERT, with some of them moving far from controls, and this suggests an accumulation of misfolded proteins." First: Proper folding of what? Second: The term "moving/moved far from controls" (which is used several times throughout the manuscript) is too informal, i. e. please better convey what is meant here (protein levels substantially differed from the corresponding values in control subjects).
- Line 479: Please delete "NK" here.
- Line 495: I propose to replace "processes" by "protein levels".
Author Response
The authors thank the Reviewer for his/her constructive comments. Point-by-point responses to the reviewer’s comments are provided below, as requested.
Reviewer 2
We would like to thank the reviewer for his fruitful observations that we have taken in full consideration while revising the manuscript. For such a reason the results section has been completely changed describing results following the Ingenuity Pathway Analysis (IPA) that we have implemented in our manuscript to improve paper comprehension and readability. We are conscious that proteomics results are difficult to digest. We hope that in this new version the paper can be more appreciated by the reader.
Remarks Introduction:
- The overall aim of the study (as I tried to summarise it further above) is not clearly depicted in the introduction. In particular, it is not yet properly addressed that this study's results might help to understand why clinical effectiveness of ERT decreases after several months, resulting in a "plateau phase" which may last for several years before progressive decline may reoccur in a subset of patients.
Author response.
We have, in this new version, addressed this question more specifically better clarifying the aim of this study in the introduction.
- It should be made more clear that the study aimed to specifically and directly compare pre-ERT and post-ERT characteristics of the muscle proteasome.
Author response.
We have made clearer the comparison pre ERT and post ERT, but we underline also that the study describes changes observed in patients comparing them to healthy age-mached subjects, which we considered a reference for two main reasons: the first was to define changes induced by the treatment on the muscle proteome, the second to find molecules that can be targeted to counteract the decline of the muscle. This can suggest, as the first reviewer pointed out, that patient’s outcome can be improved by NBS early diagnosis and by the early start of the therapy, anticipating the decline of the tissue. Muscle decline is naturally occurring with age but is exacerbated by GAA deficiency. For such a reason we address our main effort to understand how a global view of the muscle proteome can contribute and possibly to define a more personalized treatment for Pompe patients, anticipating the treatment to the disease onset.
- I am quite unhappy with the term "engulfment of lysosomes" which is used several times in a way which I think is incorrect. Engulfment is what what happens to any material intracellularly taken up by lysosomes, but this term does not properly describe what happens to the lysosomes themselves in Pompe disease or other autophagic processes, respectively. Thus, terms like "degradation", "dysfunction" or "involvement" (of lysosomes may be more appropriate).
Author response.
We have changed the term “engulfment“ with terms "degradation", "dysfunction" or "involvement" throughout the text.
- Line 83: In LOPD, there is no "unavailability of glycogen" since degradation of cytoplasmatic glycogen is not affected by the mutation.
Author response
This sentence has been amended.
Remarks Materials and methods:
- Line 542: "one" and "1" are redundant.
Author response
This redundancy has been corrected.
- Line 543: The abbreviation "PD" for Pompe disease should be avoided since it usually stands for Parkinson's disease.
Author response
We have now removed the abbreviation throughout the text.
- Line 586: The abbreviation CTR has not been introduced before, please amend.
Author response
CTR definition has been introduced.
- Line 615: The abbreviation "pI" has not yet been introduced.
Author response
pI has been defined.
- Line 638: Since both devices are manufactured by Thermo Fisher it is sufficient to mention the company once.
Author response
Corrected.
- Line 665: There is a dot between 12% and 12-18% but it should be a comma.
Author response
Corrected.
Remarks Results:
- Unfortunately, the results section is almost impenetrable. It is lacking a clear-cut structure and the vast number of abbreviations is of no help, too. I strongly recommend to thoroughly revise this section in order to improve readability. Results should be presented in a way that makes it easier to understand the facts and to grip the principles that (may) lie behind these facts. Otherwise, it is impossible to get an idea which of the observations might be meaningful (and which might not). Graphical presentation of results as it is provided in Tables 3 and 4 (with horizontal bars indicating any change from LOPD PRE to LOPD POST) is much more convenient than data presentation in Figure 2 and Table 1. It is prudent to enable the reader to get a visual impression of those changes that reach statistical significance on group comparison (LOPD PRE vs. CTR, LOPD PRE vs. LOPD POST). Regarding direct comparison between LOPD POST and CTR the respective data don't have to be presented in detail as long as they show no difference from the LOPD PRE-CTR analysis. What the reader ist interest in is any (significant) change that is likely to be caused by ERT. For graphical presentation of data, distinct proteins should be categorised as it has been already done (in subchapters 2.1.1-2.1.4) but also grouped according to the specific difference compared to CTR or LOPD PRE vs. POST. If this is carried appropriately, the large number of abbreviations may be reduced for the sake of readability.
Author response
The results section has been revised following the Reviewer suggestions introducing results adopting the strategy of IPA software analysis to improve paper readability and clarify changes based on new statistical approaches.
- Figure 2 (if it is kept): glycolytic (spelling error).
Author response
Corrected.
- Table 1 (if it is kept): Please write "%" out in the heading, i. e. "Percentage".
Author response
Deleted.
- Line 101: Please replace "have been" by "were".
Author response
Replaced.
- Line 106: Please replace "over" by "out of".
Author response
Replaced.
- Line 107: Please replace "changed" by "altered".
Author response
Replaced.
- Line 320: microscopy.
Author response
Corrected.
- Line 368: Please stick to LOPD PRE and LOPD POST and do not use PRE- and POST-LOPD
Author response
Corrected.
Remarks Discussion
- The discussion section still lacks a red thread that goes from the initial hypotheses to a concise summary of findings and to data interpretation regarding pathophysiology and clinical significance. The authors should outline which portions of the proteasome show difference as compared to CTR, which subgroups of proteins obviously "respond" to ERT and how this might translate into a), functional improvement which is partial at least, and b), persisting cellular dysfunction on the proteasome level that possibly explains why clinical effectiveness of ERT is still suboptimal or limited.
Author response
The discussion has been re-focused based on the new result section. And we have taken into full consideration the Reviewer suggestion. We hope that now this new version can overcome the constraints.
- Line 425: "FHL3" is not helpful here, please delete.
Author response
Deleted.
- Line 428: I wonder why the authors specifically relate cytoskeletal proteins and troponin to fast fibres here.
Author response
The sentence has been revised.
- Line 431: "(...) by comparing the different abundance of contractile and sarcomeric proteins in the proteome of the same patient before and after ERT (...)". Same patients, I suppose?
Author response
Yes, correct.
- Line 432: What does DMD stand for?
Author response
Dystrophin from UNIPROT database.
- Line 436: "It is well known that patients are characterized by muscle impairment which is initially ameliorated by ERT but not maintained after one year of treatment [41]." This statement is incorrect since extensive data support the clinical observation of a plateau phase in which motor function is maintained for more than a year.
Author response
We have amended this sentence and we have described more carefully the obtained results.
- Line 450: "As pointed out by proteomic analyses, the major drawback is represented by metabolism." I guess the authors want to convey that proteomic analyses indicate cell metabolism to be primarily disrupted by LOPD as reflected by the plethora of "metabolic" proteins which show altered expression. Please put this more clear-cut.
Author response
We have revised the paragraph according to Reviewer suggestions.
- Line 473: "Concerning stress proteins, a number of proteins involved in proper folding are severely dysregulated and not recovered after ERT, with some of them moving far from controls, and this suggests an accumulation of misfolded proteins." First: Proper folding of what? Second: The term "moving/moved far from controls" (which is used several times throughout the manuscript) is too informal, i. e. please better convey what is meant here (protein levels substantially differed from the corresponding values in control subjects).
Author response.
We have revised the text. Since, based on our knowledge, we could only make hypothesis about stress proteins, more specific studies are requested to investigate this issue. UPR has to be considered a hint for future studies in this direction.
- Line 479: Please delete "NK" here.
Author response.
Deleted.
- Line 495: I propose to replace "processes" by "protein levels".
Author response
Replaced.

Round 2
Reviewer 2 Report
This manuscript has had clear benefits from the revisions the authors made. However, overall readability and data interpretation still can be improved. As the authors correctly stated proteomics data are sometimes hard to digest. Thus, it is even more important to put substantial effort in clear-cut data presentation. Again, I strongly recommend to have a native speaker revise the manuscript. For further comments, see below.
Introduction
- Line 91: “Defects in GAA promote not only “depletion of readily available glycogen“. This is incorrect. Only 3% of intracellular glycogen are located within lysosomes, and in Pompe disease it’s not “unavailability” of glycogen for energymetabolism but intralysosomal accumulation that initiates the pathological cascade leading to myocyte loss.
Results
- Line 462: “… increased“. Please rephrase: ”… were increased in LOPD PRE compared with CTR.”
- Line 463: “… decreased…”: Please rephrase: “… were decreased…”.
- Line 465: “with a minute or substantial tendency toward normalization“. Please delete “a minute or”. A general point: The authors should clarify on which statistical finding they base the term “tendency”. This occurs at least twice in the manuscript. Please note, that tendency is NOT defined by narrowly missed statistical significance.
- Line 474: “were less abundant“: Please stick to the phrase “were decreased” as in the paragraphs above.
- Figures 3 and 4: There is a difference between the figures with regard to depiction of “non significant”. In Fig.3 this is depicted by dots, in Fig. 4 by grey shading of the box. Please apply the same principle of depiction in both figures, if possible.
- Line 502 and others: With regard to readability it is not helpful to start a sentence with an extensive enumeration of protein abbreviations before the reader is told by the verb what this sentence will be about or what its actual message is, respectively. Thus, try to be more straight forward with this: “When … and … were compared the following proteins were increased/decreased: … (enumeration)”.
- Figure 5. See comment above. Please harmonize the way non-significance is depicted in the figures.
Discussion
- Line 641: “Conversely, pathways analysis indicate: sirtuin signaling, RhoA signaling, glycolysis, gluconeogenesis although not significantly, as dysregulated, and PKR/interferon induction, as activated.“ This sentence is inpenetrable. Please clarify.
Conclusion:
Line 990: “Our results indicate that after 1 year of ERT, besides removing the primary cause, which still represents the ultimate treatment, targeting secondary effects could improve patients’ outcome.” It is not adequate to convey that ERT is the “ultimate” treatment of LOPD since this term ignores known limitations of the current approach including uptake and intracellular availability of the recombinant enzyme. In addition, clinical shortcomings of current ERT are evident.
Author Response
Comments and Suggestions for Authors
This manuscript has had clear benefits from the revisions the authors made. However, overall readability and data interpretation still can be improved. As the authors correctly stated proteomics data are sometimes hard to digest. Thus, it is even more important to put substantial effort in clear-cut data presentation. Again, I strongly recommend to have a native speaker revise the manuscript. For further comments, see below.
Introduction
- Line 91: “Defects in GAA promote not only “depletion of readily available glycogen“. This is incorrect. Only 3% of intracellular glycogen are located within lysosomes, and in Pompe disease it’s not “unavailability” of glycogen for energymetabolism but intralysosomal accumulation that initiates the pathological cascade leading to myocyte loss.
Author response
We thank the reviewer for the correction. We have now changed the sentence to: Defects in GAA cause lysosomal dysfunction due to glycogen overload, thus compromising cell homeostasis
Results
- Line 462: “… increased“. Please rephrase: ”… were increased in LOPD PRE compared with CTR.”
- Line 463: “… decreased…”: Please rephrase: “… were decreased…”.
- Line 465: “with a minute or substantial tendency toward normalization“. Please delete “a minute or”.
Author response
We have modified the wording as suggested, and used “were decreased” or “were increased” throughout the results section.
- A general point: The authors should clarify on which statistical finding they base the term “tendency”. This occurs at least twice in the manuscript. Please note, that tendency is NOT defined by narrowly missed statistical significance.
Author response
We now define “tendency to normalize” page 6, in brackets, as follows: “values moved closer to, but remained significantly different from controls”.
- Line 474: “were less abundant“: Please stick to the phrase “were decreased” as in the paragraphs above.
Author response
We amended the phrase as suggested
- Figures 3 and 4: There is a difference between the figures with regard to depiction of “non significant”. In Fig.3 this is depicted by dots, in Fig. 4 by grey shading of the box. Please apply the same principle of depiction in both figures, if possible.
Author response
In figures 3, 4 and 5, dots indicate non significant values for z-score (z-score ≥ 2, ≤ -2); while, in figures 4 and 5, grey color indicates changes that are non statistically significant for the p-value (ANOVA followed by Tukey’s post-hoc test (p<0.05)). Dot and grey color indicate non statistically significant values derived from two different statistics, respectively z-score and p-value, as marked down by the IPA software. We now specify this in the legends.
- Line 502 and others: With regard to readability it is not helpful to start a sentence with an extensive enumeration of protein abbreviations before the reader is told by the verb what this sentence will be about or what its actual message is, respectively. Thus, try to be more straight forward with this: “When … and … were compared the following proteins were increased/decreased: … (enumeration)”.
Author response
We apologize for the sloppiness. We have now changed most sentences according to the reviewer’s suggestions.
- Figure 5. See comment above. Please harmonize the way non-significance is depicted in the figures.
Author response
See answer to the question above
Discussion
- Line 641: “Conversely, pathways analysis indicate: sirtuin signaling, RhoA signaling, glycolysis, gluconeogenesis although not significantly, as dysregulated, and PKR/interferon induction, as activated.“ This sentence is inpenetrable. Please clarify.
Author response
We have now slightly modified this sentence so that it appears that it is used to introduce the part of discussion that is following
Conclusion:
Line 990: “Our results indicate that after 1 year of ERT, besides removing the primary cause, which still represents the ultimate treatment, targeting secondary effects could improve patients’ outcome.” It is not adequate to convey that ERT is the “ultimate” treatment of LOPD since this term ignores known limitations of the current approach including uptake and intracellular availability of the recombinant enzyme. In addition, clinical shortcomings of current ERT are evident.
Author response
We have now amended this phrase